# Effects of Sodium Selenite Injection on Serum Metabolic Profiles in Women Diagnosed with Breast Cancer-Related Lymphedema—Secondary Analysis of a Randomized Placebo-Controlled Trial Using Global Metabolomics

**DOI:** 10.3390/nu13093253

**Published:** 2021-09-18

**Authors:** Seoha Lee, Bora Lee, Yeonhee Kim, Sohyun Min, Eunjoo Yang, Seungmin Lee

**Affiliations:** 1Department of Food and Nutrition, BK21 FOUR Project, College of Human Ecology, Yonsei University, Seoul 03722, Korea; heeju0409@yonsei.ac.kr (S.L.); yh0227@yonsei.ac.kr (Y.K.); minsh@yonsei.ac.kr (S.M.); 2Graduate Program in Biomedical Engineering, College of Medicine, Yonsei University, Seoul 03722, Korea; boralee2@yonsei.ac.kr; 3Department of Rehabilitation Medicine, College of Medicine, Seoul National University Bundang Hospital, Seoul National University, Seongnam 13620, Korea; graceloves@gmail.com

**Keywords:** breast cancer-related lymphedema (BCRL), upper limb lymphedema, Selenium, sodium selenite, intravenous (IV) injection, untargeted metabolomics, global metabolomics, metabolites

## Abstract

In our previous study, intravenous (IV) injection of selenium alleviated breast cancer-related lymphedema (BCRL). This secondary analysis aimed to explore the metabolic effects of selenium on patients with BCRL. Serum samples of the selenium-treated (SE, *n* = 15) or the placebo-controlled (CTRL, *n* = 14) groups were analyzed by ultra-high-performance liquid chromatography with Q-Exactive Orbitrap tandem mass spectrometry (UHPLC-Q-Exactive Orbitrap/MS). The SE group showed a lower ratio of extracellular water to segmental water (ECW/SW) in the affected arm to ECW/SW in the unaffected arm (arm ECW/SW ratio) than the CTRL group. Metabolomics analysis showed a valid classification at 2-weeks and 107 differential metabolites were identified. Among them, the levels of corticosterone, LTB4-DMA, and PGE_3_—which are known anti-inflammatory compounds—were elevated in the SE group. Pathway analysis demonstrated that lipid metabolism (glycerophospholipid metabolism, steroid hormone biosynthesis, or arachidonic acid metabolism), nucleotide metabolism (pyrimidine or purine metabolism), and vitamin metabolism (pantothenate and CoA biosynthesis, vitamin B_6_ metabolism, ascorbate and aldarate metabolism) were altered in the SE group compared to the CTRL group. In addition, xanthurenic acid levels were negatively associated with whole blood selenium level (WBSe) and positively associated with the arm ECW/SW. In conclusion, selenium IV injection improved the arm ECW/SW ratio and altered the serum metabolic profiles in patients with BCRL, and improved the anti-inflammatory process in lipid, nucleotide and vitamin pathways, which might alleviate the symptoms of BCRL.

## 1. Introduction

Breast cancer-related lymphedema (BCRL) is a type of secondary lymphedema that stems from the disruption or obstruction of the lymphatic system after breast cancer surgery and axillary radiation therapy [1,2]. The clinical manifestation of BCRL is characterized by swelling of the upper body, especially the arms, shoulders, and necks of affected patients [3]. A meta-analysis study reported that upper limb lymphedema most frequently occurred within 2 years after a breast cancer diagnosis or surgery [4]. Another study indicated that the incidence was highest (9.8%) at 4 years, which was more than twice the incidence rate 2 years after surgery. A previous nationwide study in Denmark reported that the prevalence rate of perceived lymphedema after breast cancer treatment was approximately 65% [5]. Breast cancer survivors with BCRL reportedly experience more disabilities, lower quality of life, and higher psychological distress than survivors without BCRL [6,7,8,9].

The exact pathology of lymphedema remains unknown [10]. The sequence of development may be inflammation, progressive tissue fibrosis, worsening lymphatic functions, and finally lymphatic system disruption [11]. It has been proposed that inflammation is associated with the accumulation of adipose tissues due to damaged lymph vessels [12], which is a clinical feature of secondary lymphedema [13]. It has also been proposed that inflammation, along with adipose tissue hypertrophy and increased adipocyte numbers, may impair lymph vessels in an obese mouse model [14]. Additionally, T-helper 2 cell differentiation significantly suppressed lymphedema by decreasing tissue fibrosis and improving lymphatic function in a T-cell deficient mouse model [10], providing further evidence for the relationship between inflammation and lymphedema.

Several genes are associated with lymphedema development and progression [15]. Mutations in vascular endothelial growth factor C (VEGFC) and vascular endothelial growth factor receptor 3 (VEGFR3) induce altered lymph formation, which may progress to lymphedema [16]. In particular, overexpression of VEGFC resulted in lymphedema exacerbation accompanied by higher immune cell infiltration and vascular leakage in a mouse model [17]. In addition, NFKB2 and FOXC2 have been proposed to be primary factors in lymphangiogenesis and lymphedema since they are involved in both inflammatory reactions and lymph development [1]. Tyrosine kinase with immunoglobulin-like and EGF-like domains 1 (TIE1)—a cell surface protein involved in angiogenesis and lymphangiogenesis—is reportedly related to lymphatic dysfunction, as TIE1 gene variants were identified in 235 lymphedema patients [18]. The TIE1 receptor-deficient mouse model exhibited abnormal development of lymphatic vessels [18], indicating that the TIE1 gene may be associated with lymphedema progression.

Since no clear treatment has been established for lymphedema [19], disease management focuses on ameliorating symptoms, including reducing swelling in the affected limb and decreasing the risk of recurrent infections [20]. Pharmacological interventions, including benzopyrones, flavonoids, and corticosteroids, are used to reduce the proportion of free radicals and decrease the limb circumference in lymphedema patients [21]. Benzopyrones, the most commonly used drug in lymphedema treatment, significantly reduce lymphedema [21]. Ketoprofen, a non-steroid anti-inflammatory drug, reduced tail volumes in mice with acquired lymphedema by promoting VEGFC-induced lymphangiogenesis [22]. In addition, tacrolimus, an anti-T-cell agent, inhibited lymphangiogenesis and decreased fibrosis in a mouse model [23].

Selenium (Se) has been suggested as a therapeutic agent for lymphedema because of its anti-edema role in several studies [2,21,24,25]. Kasseroller et al. [2] demonstrated that oral administration of sodium selenite elevated whole blood Se levels (WBSe) and reduced lymphedema volume compared to the placebo. Micke et al. [24] reported that the administration of selenium in patients with lymphedema after radiotherapy improved one or more clinical lymphedema stages and reduced swelling. Zimmermann et al. [25] showed that Se intervention reduced lymphedema volume in patients who underwent oral tumor surgery in a double-blind, randomized prospective study. Accordingly, studies consistently suggest that Se may clinically improve lymphedema, but little is known about the mechanism [21]. Se activates endogenous antioxidative compounds such as glutathione peroxidase (GPx) [24,25], an enzyme that reduces excessive free radicals produced in lymphedema tissues [26]. Lymphedema severity was positively correlated with reactive oxygen species levels and negatively correlated with WBSe and GPx [25]. Moreover, Se may help alleviate lymphedema by reducing inflammation by disturbing nuclear transcription factor κB (NF-κB), which binds to human T cell DNA [27]. Se-induced enhancement of the immune system may have a positive effect on lymphedema by decreasing edema volume and the incidence of lymphedema complication [24,28,29]. Se-intervention in head and neck cancer patients significantly increased cellular immune responses, including cytotoxic lymphocytes and other immunologic parameters [28]. Our previous study [30] examining the antioxidative effects of selenium as anti-BCRL revealed that the improvement observed in patients with BCRL was not due to the antioxidative activity of Se. Therefore, other mechanistic aspects of the anti-edema effects of Se warrant further studies on its mechanism of action.

Global metabolomics is an unbiased and effective method to reveal biological changes underlying relevant events [31]. Global metabolomics research on the association between selenium and BCRL may reveal an unbiased metabolic mechanism of action of anti-edemic effects of selenium in the BCRL since metabolite profiles depict detailed molecular compositions of tissue or serum samples [32]. Previously, lipid profiling analysis in the adipose tissues of patients with both primary and secondary lymphedema showed that lymphedema may exhibit chronic low-grade inflammation with increased levels of pro-inflammatory lipids such as arachidonic acid and ceramides [33]. In a recent omics study, dietary Se deficiency significantly altered various metabolites including amines, sugars, organic acids, purines, pyrimidines, lipids, free fatty acids, bile acids and lipid mediators [34]. It also affected redox homeostasis via the metabolism of methionine in mouse brain and liver [34]. Moreover, supplementation with Se and coenzyme Q_10_ in healthy elderly participants caused a significant metabolic shift in the pathways related to pentose phosphate, mevalonate, beta-oxidation, and xanthine oxidase, leading to reduced oxidative stress and inflammation [35]. However, a global metabolome profile analysis of patients with lymphedema after Se intervention has not been conducted to date. With these aspects, we used global metabolomics for analyzing the serum metabolomic profiles of Se-treated (SE) and placebo-controlled (CTRL) groups, expecting to find evidence of the potential mechanism related to the effects of Se IV injection on BCRL.

## 2. Materials and Methods

### 2.1. Serum Samples for the Study

Serum samples and written consent were obtained in our previous clinical study of 29 participants who exhibited severe lymphedema (stage II–III) after breast cancer therapy. That study was approved by the Institutional Review Board (IRB) of Seoul National University Bundang Hospital (approval number: 02-2012-062) [25]. In our current study, we included previously excluded outliers to increase the strength of the metabolomics analysis. Finally, 15 participants were included in the SE group and 14 in the CTRL groups.

### 2.2. Study Design

The intervention protocol used in this study was previously described in detail [25]. Briefly, participants in the SE group received 500 µg of sodium selenite (Selenase^®^, 10 mL, Boryung Pharmaceutical Co., Ltd., Seoul, Korea) diluted with 50 mL of 0.9% normal saline via IV injection five times within 2-weeks from the day of enrollment. Participants in the CTRL group received normal saline instead of sodium selenite. Serum samples were obtained before the first injection of sodium selenite (baseline), immediately after the intervention (2-weeks), and approximately 4 weeks after the last injection (follow-up). The specific time point for the 2-weeks sample collection was determined to be 1.8 ± 3.8 days after the last session and for follow-up it was determined to be 31.1 ± 8.8 days after the 2-weeks session.

### 2.3. Anthropometry, BIA, and WBSe

The ratio of extracellular water to segmental water (ECW/SW) was used instead of extracellular water to total body water (ECW/TBW). All the data regarding body weight, BMI, arm ECW/SW ratio, single-frequency bioimpedance analysis (SFBIA) ratios, and WBSe of the patients were obtained from our previous report [30]. However, the arm ECW/SW ratio was defined in our study according to the equation developed by Kim et al. [36].

$$\mathrm{Arm}\;\mathrm{ECW}/\mathrm{SW}\;ratio=\frac{{\left(\mathrm{ECW}/\mathrm{SW}\right)}_{affected\;arm}}{{\left(\mathrm{ECW}/\mathrm{SW}\right)}_{unaffected\;arm}}.$$


### 2.4. Serum Preparation for LC-MS/MS Analysis

Serum samples that were stored at −80 °C were used for liquid chromatography-tandem mass spectrometry (LC-MS/MS) analysis. For sample preparation, 100 μL of the sera, 800 μL of 70% methanol, and 10 μL of the internal standard were briefly vortexed and then left on ice for 10 min. The sample mixture was then obtained by centrifugation at 10,000 rpm for 5 min at 4 °C, and the supernatant was lyophilized overnight at −84 °C. Then, 100 μL of 10% methanol was added to the freeze-dried supernatant, vortexed, and centrifuged at 10,000 rpm for 5 min at 4 °C. The supernatant (90 μL) was stored at 4 °C until further analysis. For the internal standard, 10 μg/mL of reserpine, acetaminophen, sulfadimethoxine, and terfenadine (Sigma-Aldrich, Oakville, ON, Canada) solutions were prepared in 70% acetonitrile in autoclaved water. Then, the same volumes of each solution were mixed well and stored at 4 °C until sample preparation.

### 2.5. LC-MS/MS Analysis

Metabolic profiling was conducted on the sera of patients with BCRL who received Se or normal saline IV injections. Liquid chromatographic analysis was performed using the Ultimate 3000 UHPLC (Thermo Fisher Scientific, Waltham, MA, USA) coupled with a column (2.1 × 150 mm, C18) under the following conditions: the mobile phases were composed of 0.1% formic acid in water (A) and 0.1% formic acid in methanol (B). The gradient (%) was changed from 100/0 to 0/100 for 15 min, maintained at 0/100 for 4 min, and then changed back to 100/0 for 2 min. The flow rate was 0.4 mL/min.

MS/MS analysis was performed using a Q-Exactive Orbitrap Plus mass spectrometer (Thermo Fisher Scientific, Waltham, MA, USA). The electrospray ionization (ESI) source was operated in positive mode. MS data were collected in full scan mode (resolution: 70,000; scan range: 80–1000 m/z) and data-dependent MS^2^ (dd-MS^2^) (resolution: 17,500; Top 10). The capillary temperature and auxiliary gas heater temperature were set to 320 °C and 300 °C, respectively.

### 2.6. Data Processing and Identification of Metabolites

The raw LC-MS data were imported into the XCMS online platform (xcmsonline.scripps.edu), which is used for data processing, including peak detection, peak alignment and extraction of peak intensities, m/z, and retention time [33]. The bandwidth and tolerance were set to 10 s and 15 ppm, respectively. Other XCMS parameters were set to the default values. The differential peaks were identified by matching the exact masses and intensities in the most widely used online databases, including the MyCompoundID MS/MS Search (MCID) (www.mycompoundid.org (accessed date: 27 August 2020)) and the human metabolome database (HMDB) (www.hmdb.ca (accessed date: 27 August 2020)), using a fit score ≥ 0.7. MetaboAnalyst 4.0 (www.metaboanalyst.ca (accessed date: 27 August 2020)) was used for further statistical and functional analyses, including partial least squares-discriminant analysis (PLS-DA), orthogonal partial least squares discriminant analysis (OPLS-DA), heatmap, and pathway analysis. MetaboAnalyst 4.0 was designed to permit comprehensive metabolomics data and statistical analyses, as well as the visualization and interpretation of the metabolomics data [34].

### 2.7. Pathway Analysis

For further pathway analysis using MetaboAnalyst 4.0, a variable importance in the projection (VIP) values of >1.0, and *p* < 0.05, were used as cut-offs to identify significant metabolites. VIPs were obtained from SIMCA 16.0 (Umetrics, Göttingen, Germany). The false discovery rate (FDR) was automatically adjusted using MetaboAnalyst 4.0 to validate the significance of the metabolic pathway. The Kyoto Encyclopedia of Genes and Genomes (KEGG) database (www.kegg.jp/kegg/pathway.html (accessed date: 13th, November, 2020)) and the Small Molecule Pathway Database (SMPDB) (www.smpdb.ca (accessed date: 13 November 2020)) were used to search for the superpathways and pathways of differential metabolites.

### 2.8. Statistical Analysis

Statistical analyses, including paired t-tests, chi-squared tests, two-way ANOVA, Mann-Whitney *U*-tests and Wilcoxon signed-rank tests, were performed using the IBM SPSS Statistics v.25 software. Differences between baseline and 2-weeks or follow-up time points were analyzed using the Wilcoxon signed-rank test. Univariate nonparametric Mann-Whitney *U*-tests were performed to compare WBSe between the CTRL and SE groups for all metabolites. MetaboAnalyst 4.0 was used to conduct multivariate PLS-DA and OPLS-DA for all groups. Metabolic peak intensities were log-transformed and Pareto-scaled prior to the multivariate analysis. Log transformation was used to normalize the skewed distribution of the metabolite intensity values [36]. Pareto scaling adjusts the relative importance of large values [37]. The robustness and validity of the results were assessed with the cumulative parameters for goodness of fit (R^2^), goodness of prediction (Q^2^), and the permutation test. Metabolites with VIP values > 1.0 and univariate statistical *p*-values < 0.05 were filtered for further analysis. We performed regression analysis using a linear mixed-effect model (LMM) (R v.4.0.5, http://cran.r-project.org (accessed date: 20 March 2021)) with participants as random effects, and other variables, including group, time and arm ECW/SW ratio or the WBSe, as fixed effects in the statistical models.

## 3. Results

### 3.1. Reduced Arm ECW/SW Ratio after Se IV Injection and the Negative Correlation with WBSe in Patients with BCRL

First, we re-analyzed the general and surgical characteristics of the participants before performing the metabolomics analysis (Appendix A). No statistical significance in the factors, such as the proportions of normal weight, overweight, and obese participants, the ages, and the post-surgery time (years), were identified in comparing the CTRL and SE groups. In addition, two-way ANOVA demonstrated that overweight/obesity had no interaction with the clinical improvements of lymphedema in this study (F = 0.011, *p* = 0.917). Consequently, the possible relevant factors for lymphedema, such as ages, BMIs and post-surgery dates, were not included in our metabolomics analysis.

The efficacy of Se IV injection was re-estimated owing to the inclusion of additional participant samples. Se resulted in an increased WBSe, but showed no effects on body weight, BMI, and SFBIA ratios, consistent with our previous study [30]. In accordance with Han et al. [30], the SE group exhibited significant clinical improvement of BCRL, as the SE patients in stage 3 decreased to stage 2 by 60.0% between baseline and at 2-weeks, and by 13.4% between 2-weeks and follow-up (Appendix A).

The arm ECW/SW ratio was significantly reduced at the 2-weeks and follow-up time points compared with baseline in the SE group, whereas no difference was detected in the CTRL group (Figure 1a). In the correlation analysis, the arm ECW/SW ratio was negatively correlated with WBSe (Figure 1b) and positively correlated with the 1 kHz (*r* = 0.764), 5 kHz (*r* = 0.722), and 50 kHz (*r* = 0.866) SFBIA ratios (Figure 1c). In contrast, there were no significant correlations between the WBSe and SFBIA ratios or BMI (Figure 1c). Overall, Se IV injection reduced the arm ECW/SW ratio, which correlated negatively with WBSe and positively with SFBIA ratios.

### 3.2. Metabolomic Model Validation

A total of 14,764 ionized compounds were first obtained and processed from the UHPLC-MS/MS data to establish valid metabolic models (Figure 2a). Partial least squares-discriminant analysis (PLS-DA) was used to visualize the overall variation of the entire group, which showed that the top three components accounted for 45.8% of the total variance (Figure 2b). The Q^2^ and R^2^ of the model were greater than 0.5, and 0.9, respectively, clearly demonstrating that the groups were separated by the peaks in the PLS-DA model.

To evaluate the validity of the model between the two groups, orthogonal PLS-DA (OPLS-DA) was performed. The valid model was established between the CTRL and SE groups at 2-weeks (Table 1). Within-group comparisons showed that no valid model was constructed between baseline, 2-weeks, and follow-up time points in the CTRL as well as in the SE group, and baseline profiles of the CTRL and SE groups were not significantly different (Table 1). A clear separation between the groups was further verified by the OPLS-DA score plot, S-plot, and permutation test (Figure 2c–e). The valid models were confirmed at 2-weeks between the CTRL and SE groups.

### 3.3. Differential Metabolites and Pathways Affected by Se IV Injection Compared to the Placebo Control

A total of 107 differential metabolites were detected after excluding metabolites that differed at baseline, duplicates, drugs, and xenobiotics (Figure 2a). A heatmap of the metabolites shows the overall differences observed between the CTRL and SE groups at 2-weeks (Figure 3a). The metabolites were categorized into four main classes: (i) lipids and lipid-like molecules; (ii) organic acids and derivatives; (iii) organic heterocyclic compounds; and (iv) nucleosides and nucleotides (Figure 3b and Figure 4, Appendix A). Extensive changes in 19 pathways (*p* < 0.05), including lipid, nucleotide, and vitamin metabolism, were investigated (Figure 5a and Appendix A) in the pathway analysis.

Among the 44 lipid and lipid-like metabolites that were most abundant, 22 were classified as glycerophospholipids (GPLs) or their major intermediates—glycerolipids (Figure 3b). Of the GPLs, phosphatidylcholine (PC) (20:0/22:2) and lysoPC (20:0 and 20:1) were increased in the SE group (Figure 4 and Figure 5c, Appendix A). In addition, some phosphatidylethanolamine (PE) levels were different in the SE group (Figure 4 and Figure 5, Appendix A). Two fatty acyls involved in arachidonic acid metabolism—leukotriene B_4_ dimethylamide (LTB_4_-DMA) and prostaglandin E_3_ (PGE_3_)—were relatively high in the SE group (FDR = 0.043 and *p* = 0.0286). In the steroid and steroid derivatives class, steroid hormone biosynthesis metabolites corticosterone and dihydrocortisol were elevated in the SE group (FDR = 0.024 and *p* = 0.0061).

Vitamin B_3_ (nicotinic acid), vitamin B_5_ (pantothenic acid), vitamin B_6_, vitamin B_12_, and vitamin C (ascorbate and aldarate) were significantly different in the SE group (Figure 5b). Quinolinic acid (QA) and nicotinamide mononucleotide (NMN), which are involved in nicotinate and nicotinamide metabolism, were increased in the SE group (FDR = 0.006 and *p* = 0.0004). PPC (pantothenate and CoA biosynthesis; FDR = 0.040 and *p* = 0.0184), pyridoxamine (vitamin B_6_ metabolism; FDR = 0.043 and *p* = 0.0267), and alpha-ribazole (related to vitamin B_12_) and L-gulonolactone (also known as reduced ascorbic acid), which are involved in vitamin C metabolism (FDR = 0.021 and *p* = 0.0030), were all elevated in the SE group.

### 3.4. Metabolites Associated with WBSe or the Arm ECW/SW Ratio

To further investigate which metabolites among those altered in the SE group were associated with WBSe levels or the arm ECW/SW ratio, regression analysis was performed. The only metabolite associated with WBSe levels at 2-weeks was xanthurenic acid (XA), which showed a negative relationship (Figure 6b, Appendix A). Five metabolites were significantly associated with the arm ECW/SW ratio; positive associations were found with XA and 24,25-dihydroxyvitamin D_3_ and negative associations with hydroxyphenylacetylglycine, alpha-ribazole, and 2-methyl-3-hydroxy-5-formylpyridine-4-carboxylate (Figure 6c–g). Among the differential metabolites found in the SE group as compared to the CTRL, XA was also found to be related to WBSe levels and the arm ECW/SW ratio.

## 4. Discussion

BCRL is one of the most common complications following breast cancer surgery. The beneficial effects of Se as a treatment for BCRL have been demonstrated in several clinical studies, including our previous research [2,24,25,30]. However, the mechanism by which Se affects BCRL remains poorly understood. Here, we report for the first time altered metabolites and relevant pathways underlying the effects of Se on patients with BCRL through serum metabolic profiling analysis.

ECW/TBW has been suggested to be an edema index, representing the fluid volume status [37]. However, others reported that ECW/TBW showed no differences between BCRL and non-BCRL patients in either sentinel lymph node biopsy (SLNB) or axillary lymph node dissection (ALND) groups [38]. It is still unclear whether the ECW/TBW value represents the severity of lymphedema. In our study, no differences in ECW/TBW, affected arm ECW/SW, or unaffected arm ECW/SW were previously observed between the SE and CTRL groups at 2-weeks, despite the conspicuous progression of clinical lymphedema stage [30]. Here, we found that the arm ECW/SW ratio (affected to unaffected arm) was significantly reduced in the SE group and was inversely correlated to Se levels in the blood. In support of our finding, a significant positive correlation between the arm ECW/SW ratio and the volume ratio of each arm was reported in patients with BCRL [36]. The arm ECW/SW ratio might be a better indicator than ECW/TBW of the status of patients with BCRL. Furthermore, we found that the arm ECW/SW ratio was positively correlated with the SFBIA ratios. The SFBIA ratios have been used to assess lymphedema symptoms in several studies [38,39,40]. A higher SFBIA ratio was associated with greater differences in arm circumference [39]. Although no impact on SFBIA was detected after Se IV injection in BCRL patients, the Se injection appeared to improve arm ECW/SW ratio, suggesting that it may be an effective symptom reliever for BCRL. The close correlation of the arm ECW/SW ratio with SFBIA ratios may also suggest the potential use of the arm ECW/SW ratio as a sensitive indicator of lymphedema.

Se-induced changes in metabolite levels may contribute to the alleviation of lymphedema. Our metabolome analysis identified a total of 107 metabolites associated with Se IV injection in patients with BCRL. Among the differential metabolites detected, corticosterone, LTB4-DMA and PGE_3_ were elevated in the SE group and may be related to SE-mediated anti-BCRL effects, partly through anti-inflammatory roles. Se-rich polysaccharides were reported to increase serum corticosterone levels in a chronic fatigue rat model [41], consistent with our Se-mediated elevation of corticosterone. In another rodent model, elevation of corticosterone inhibited paw edema and chronic inflammatory responses [42]. In addition, elevation of LTB4-DMA due to SE might also play a role in SE-mediated anti-BCRL effects. LTB4-DMA is an antagonist of LTB4 [43], which is a derivative of LTB synthesized from arachidonic acid. LTB4 is reportedly involved in the molecular pathogenesis of lymphedema [44]. Antagonizing LTB4 repaired the lymph system and inflammation in the mouse tail [45]. PGE_3_, another SE-altered metabolite, has been proposed to have anti-inflammatory functions [46]. PGE_3_ is derived from EPA (n-3 fatty acid), while PGE_2_ is derived from arachidonic acid (n-6 fatty acid), which has been detected as a causal factor in lymphedema [44].

Moreover, as Se supplementation significantly changed vitamin metabolism, based on the feces of a mouse model [47], SE injection might improve BCRL by affecting metabolites involved in vitamin pathways. In particular, vitamin B_6_ has been suggested to participate in the delivery of Se from serum to tissues and to influence the biopotency of Se [48]. Among the detected metabolites involved in vitamin pathways, NMN—a precursor to NAD^+^ in vitamin B_3_ metabolism—and types of pyridoxamine in vitamin B_6_ metabolism may help treat BCRL due to their anti-inflammatory effects. NMN administration lowered levels of Il1b, an inflammatory cytokine, in pancreatic islets from a fructose-rich diet-fed mouse model [49]. In addition, various metabolites involved in vitamin B_6_ metabolism suppress inflammatory cytokines including IL-6, TNFα, and IL-1β [50,51,52]. Thus, NMN and pyridoxamine, which were elevated in the SE group, may have anti-inflammatory effects that result in anti-edema responses in BCRL patients.

Our pathway analysis further identified alterations in metabolites with relatively high relevance to glycerophospholipid metabolism, pyrimidine metabolism, and purine metabolism. Considering that PC decreased the prevalence and symptoms of lymphedema in a rat model [53], the elevations in various types of PCs in the SE group might also contribute to the beneficial effects of Se in patients with BCRL. In a previous study, choline administration increased the PC/PE ratio, reduced inflammation, and increased the survival rate in high-fat diet-fed mice [54]. The elevation of PCs seen in the SE group could lead to an increased ratio of PC/PE and might lower the inflammatory response. UMP, a metabolite of pyrimidine metabolism, increased in the SE group and has been reported to increase brain concentrations of CDP-choline (cytidine diphosphate-choline), which is a precursor of PC [55]. As CDP decreased paw edema in an acute inflammatory pain rat model [56], UMP seen in the SE group might have elevated CDP-choline, which in turn might participate in the alleviation of lymphedema. UDP, also altered in the SE group, exhibited a significant reduction in neurologic abnormalities when applied to rabbits with cold injury-induced brain edema [57]. Therefore, elevated metabolites involved in pyrimidine metabolism, such as UMP and dUDP, might have anti-edema effects in patients with BCRL. The decrease of AMP in the SE group might help treat lymphedema due to its involvement in angiogenesis, which has been suggested as a cause of lymphedema [58]. Certain concentrations of AMP have been found to be key activators of AMP-activated protein kinase (AMPK) [59]. One study found that promotion of AMPK was significantly associated with increased generation of VEGF, leading to angiogenesis in vivo [60]. Overall, our findings may provide evidence that changes in glycerophospholipid metabolism, pyrimidine metabolism, and purine metabolism partly help to improve lymphedema in patients with BCRL, even though the effects of Se on these alterations warrant further validation.

Other metabolites in several pathways, including vitamin B_12_, ascorbate and aldarate, cysteine and methionine and tryptophan, might also affect BCRL by reducing inflammation. Vitamin B_12_ showed a negative relationship with TNF-α, a pro-inflammatory indicator, in the sera of healthy participants [61]. Alpha-ribazole, which showed a positive relationship with the arm ECW/SW ratio in our statistical data, is involved in the biosynthetic pathway of adenosylcobalamin, a coenzyme of vitamin B_12_ [62]. Thus, elevations in alpha-ribazole might help to reduce inflammation in BCRL patients. Vitamin C suppresses TNFα-induced NF-κB activation in vitro [63,64]. In addition, L-gulonolactone has been suggested to reverse edema in both animals and humans [65,66]. Our analysis also showed a positive relationship between L-gulonolactone and the arm ECW/SW ratio, indicating that Se’s anti-edema effect might be related to L-gulonolactone. In addition, SAM, which is involved in cysteine and methionine metabolism, was also increased in the SE group and has been shown to decrease the expression levels of inflammatory cytokines, including TNF-α, while increasing anti-inflammatory cytokines such as IL-10 [67].

Of note, XA might directly relate the changes in WBSe to the clinical status of patients with BCRL, as it was commonly identified in association with both WBSe and the arm ECW/SW ratio. Previously, sodium selenite supplemented mice showed the suppressed generation of immunomodulators such as XA compared with Se-deficient mice [47], implicating the potent impact of Se on the decreased levels of XA detected in our study. XA is produced upon the degradation of tryptophan, which is activated by indoleamine-2,3-dioxygenase (IDO) in the kynurenine pathway of tryptophan metabolism [68]. The activity of IDO and IDO-related metabolism has been suggested to be more active with inflammation [69]. In addition, higher XA levels were found in the plasma of patients with type 2 diabetes than in those without diabetes [70]. Thus, WBSe-diminished XA may have decreased inflammation in our patients, which in turn led to a decrease in the arm ECW/SW ratio. The serum levels of XA may act as a candidate marker for BCRL, especially in response to Se injection.

In addition, based on our regression analysis of the arm ECW/SW ratio, BCRL might be relieved by higher levels of hydroxyphenylacetylglycine and 2-Methyl-3-hydroxy-5-formylpyridine-4-carboxylate, and lower levels of 24, 25-dihydroxyvitamin D_3_. 24, 25-dihydroxyvitamin D_3_ was shown to inhibit 1, 25-dihydroxyvitamin D_3_ [71], which decreases inflammation by decreasing NF-κB [72].

In conclusion, we explored the possible mechanisms by which Se treats BCRL using a global metabolomics analysis. Some potential anti-inflammatory benefits of Se IV injection in BCRL patients are suggested in Figure 7. However, the main limitation of this study is that we did not seek clinical evidence on the anti-inflammatory effects of Se. Because this was a secondary analysis, additional research to confirm our findings on the potential anti-inflammation by Se was not allowed. Therefore, there is a critical need for efforts toward unveiling the exact mechanism of the anti-inflammatory effects of Se. 

## Figures and Tables

**Figure 1 nutrients-13-03253-f001:**
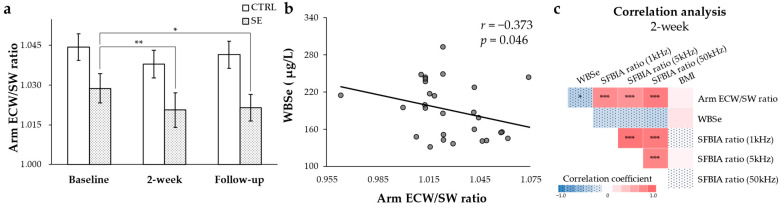
Arm ECW/SW ratio and the correlation between the arm ECW/SW ratio, WBSe, SFBIA ratios, and BMI. (**a**) Changes in the arm ECW/SW ratio at three time points (baseline, 2-weeks, and follow-up). The bar graph represents the mean ± standard error. (**b**) Negative correlation between the arm ECW/SW ratio and WBSe. (**c**) Matrix of the Spearman’s correlation coefficients between the WBSe, arm ECW/SW ratio, SFBIA ratios, and BMI at 2-weeks. * *p* < 0.05; ** *p* < 0.01; *** *p* < 0.001.

**Figure 2 nutrients-13-03253-f002:**
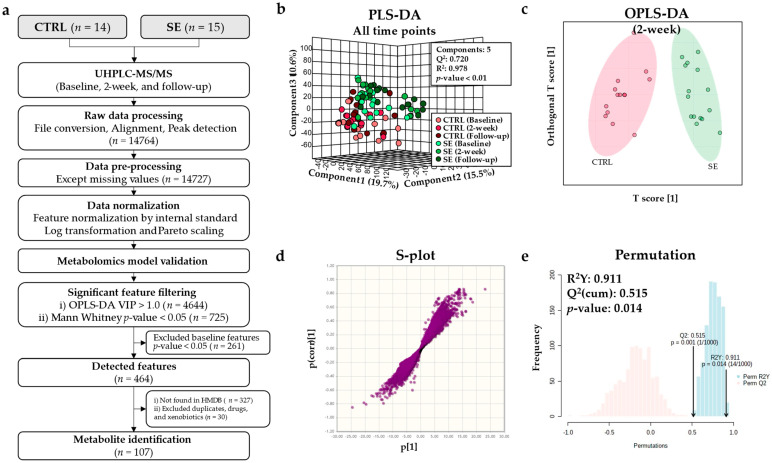
Modeling for the metabolite analysis. The raw intensity metabolite data were normalized using the internal standard, Pareto scaling, and log transformation. (**a**) Flowchart of the metabolomics study. (**b**) Partial least squares-discriminant analysis (PLS-DA) three-dimensional scaled score plot at all time points. (**c**–**e**) Metabolic profiling results for the placebo−controlled (CTRL) and Se-treated (SE) groups at 2-weeks. (**c**) Orthogonal partial least squares-discriminant analysis (OPLS-DA) score plot. (**d**) S-plot. (**e**) Permutation test.

**Figure 3 nutrients-13-03253-f003:**
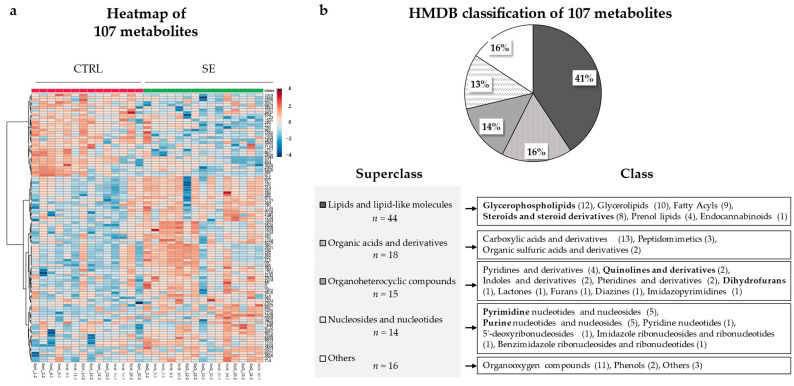
Differential metabolites between the CTRL and SE group at 2-weeks. A total of 107 metabolites were detected by a variable importance in the projection (VIP) > 1.0 in OPLS-DA and *p* < 0.05 in the Mann-Whitney *U*-test between the groups. The raw intensity data of metabolites were normalized using the internal standard, Pareto scaling, and log transformation. (**a**) Heatmap visualization of the intensities of the differential metabolites. The stratified color from red to blue represents the respective increase or decrease in relative intensity (**b**) Classification of metabolites into *Superclass* and *Class* according to their biological role based on the human metabolome database (HMDB). The number of metabolites in each *Class* are represented in parentheses.

**Figure 4 nutrients-13-03253-f004:**
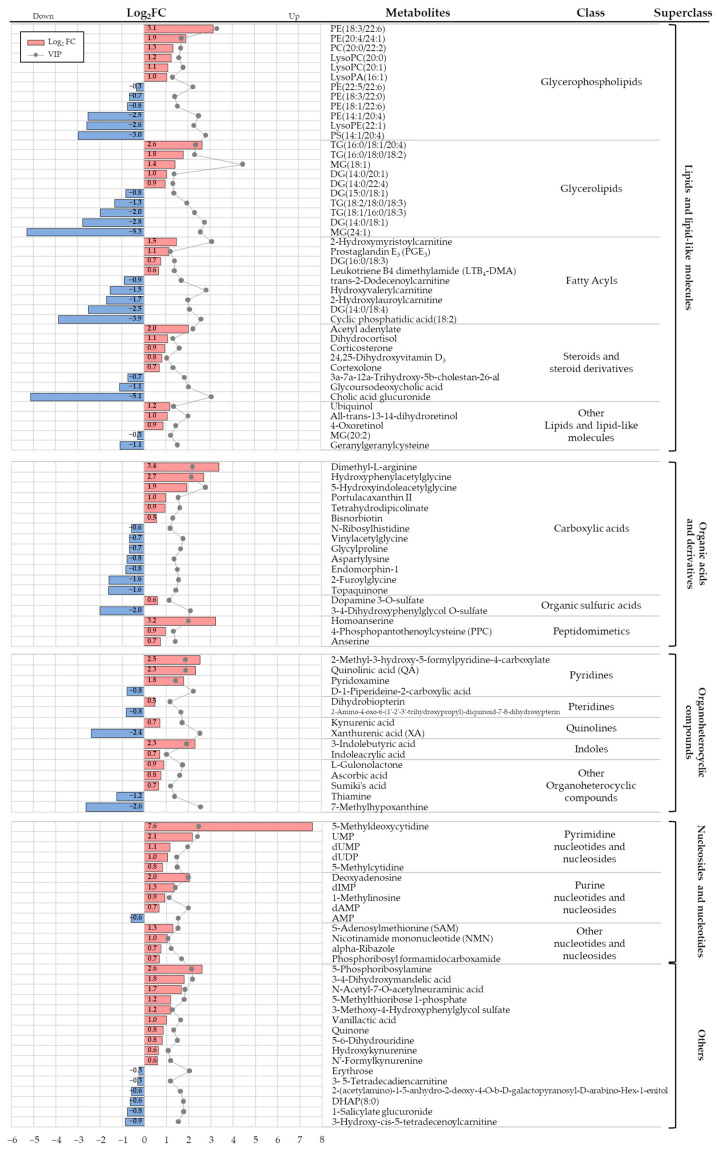
The relative abundance of the differential metabolites. The log_2_FC (fold change) indicates the relatively high or low levels of intensities of the significant metabolites in the SE group compared to the CTRL. The red and blue colors indicate the log_2_FC values above and below 0, respectively. Metabolites were arranged in the order of the log_2_FC values, from high to low.

**Figure 5 nutrients-13-03253-f005:**
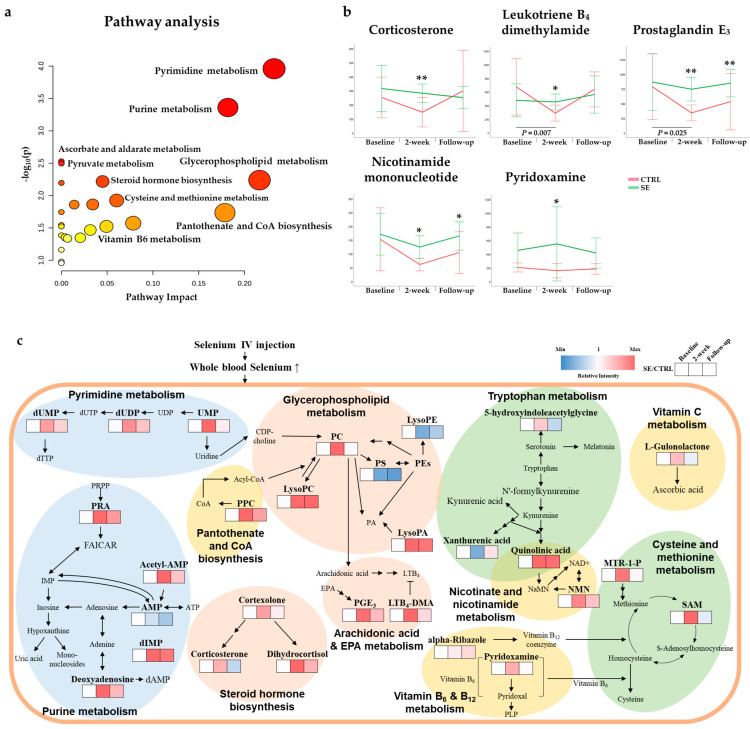
Altered metabolites and metabolic pathways. (**a**) Pathway analysis based on the identified metabolites. (**b**) Box plots of the five differential metabolites which are possibly involved in anti-inflammatory mechanisms. Significance of group comparisons at specific time points is denoted with an asterisk (* *p* < 0.05 and ** *p* < 0.01), while raw *p*-values represent significant changes compared to baseline in the CTRL group. (**c**) Altered serum metabolic profiles concerning Se IV injection in the patients with BCRL. Significantly changed metabolites are represented in bold. Stratified red to blue color shows baseline-adjusted relative intensity of the SE group compared to the CTRL, from high to low.

**Figure 6 nutrients-13-03253-f006:**
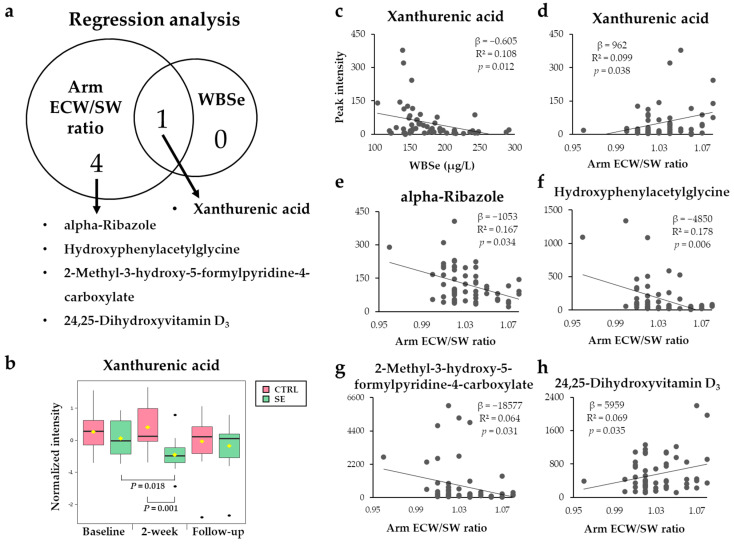
Associations of differential metabolites with WBSe or the arm ECW/SW ratio. (**a**) Venn diagram showing metabolites significantly associated with WBSe and the arm ECW/SW ratio. (**b**) Box plot of xanthurenic acid, which was related to both WBSe and arm ECW/SW ratio. (**c**–**h**) Linear regressions representing intensity distribution of metabolites significantly associated with WBSe (**c**) or the arm ECW/SW ratio (**d**–**h**).

**Figure 7 nutrients-13-03253-f007:**
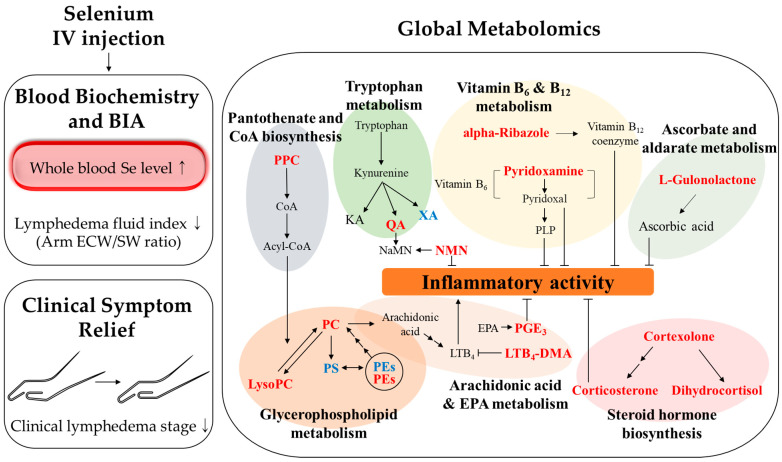
Proposed mechanism of Se IV injection in the serum of the patients with BCRL.

**Table 1 nutrients-13-03253-t001:** Quality assessment parameters and permutation test for model validation of OPLS-DA to distinguish the CTRL and SE groups.

Group Comparisons ^2^	OPLS-DA ^1^			
	Q^2^ (cum)	R^2^ (cum)	Q^2^ (cum)/R^2^ (cum)	Permutation Test (*p*-Value)
Between groups (CTRL–SE)				
0–0	0.543	0.988	0.550	0.21
**2–2**	**0.515**	**0.911**	**0.565**	**0.01**
6–6	0.509	0.987	0.516	0.14
Within groups (CTRL–CTRL)				
0–2	0.335	0.813	0.412	0.99
0–6	0.203	0.993	0.204	0.80
2–6	0.113	0.910	0.124	0.29
Within groups (SE–SE)				
0–2	−0.309	0.996	−0.310	0.23
0–6	−0.104	0.911	−0.114	0.01
2–6	−0.325	0.691	−0.470	0.59

^1^ OPLS-DA, orthogonal partial least squares discriminant analysis. ^2^ Time points were compared between and within groups and defined as: 0, baseline; 2, 2-weeks; 6, Follow-up. E.g., ‘0–0’ under CTRL–SE represents CTRL at baseline vs. SE at baseline; ‘0–2’ under CTRL–CTRL represents CTRL at baseline vs. CTRL at 2-weeks. The valid model is represented in bold.

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
