# Peer review of "Effects of Sodium Selenite Injection on Serum Metabolic Profiles in Women Diagnosed with Breast Cancer-Related Lymphedema—Secondary Analysis of a Randomized Placebo-Controlled Trial Using Global Metabolomics"

_nutrients, 2021, doi:10.3390/nu13093253_

Round 1
Reviewer 1 Report
The manuscript is based on a secondary analysis on the effects of intravenous (IV) administration of selenium as a new treatment to treat breast cancer-related lymphedema (BCRL).
lymphedema caused by breast cancer (BCRL). In this study the authors have analyzed the metabolic effects of selenium in serum samples from patients in the selenium (SE, n = 15) or placebo control group 18 (CTRL, n = 14) were analyzed by ultra-high performance liquid chromatography with UHPLC-Q-Exactive Orbitrap/MS. The topic is more related to oncology and pharmacology fields than to Nutrients in my opinion. The number of samples is quite low and the authors should explain why they have not analyzed all samples from the previously published clinical trial.
There is a quite large consensous about the role of overweight/obesity on lymphedema ocurrence after breast cancer as summarized in the following meta-analysis:
Wu R, Huang X, Dong X, Zhang H, Zhuang L. Obese patients have higher risk of breast cancer-related lymphedema than overweight patients after breast cancer: a meta-analysis. Ann Transl Med. 2019 Apr;7(8):172.
Thus it is clinical relevant to analysed the changes in metabolic profiles based on overweight/obesity as a confounding factors. In fact no signficant confounding factors have been analysed in the study such the age of the patients enrolled in the clinical trial or the effect of previous chemotherapy administration. tHis is crucial because these factors can alter metabolic profile and bias in some how at least some of the metabolic effects induced by selenium administration.
Since the authors suggest (often in the discussion) that many of the metabolic alterations observed after selenium administration are explained by its anti-inflammatory effects, the authors should measure in blood classical inflammatory markers that are measured in routine clinical practice such as C-reactive protein, ferritin, white blood cell count or erythrocyte sedimentation rate for example. These measurements could corroborate whether the metabolic changes correspond to an anti-inflammatory effect of selenium.
Reviewer 2 Report
nutrients-1337185
In this manuscript, the authors investigated the effect of selenium on serum metabolites on BCRL patients. Selenite injections show a significantly decrease in arm ECW/SW ratio in which negatively linked to blood selenium levels and arm ECW/SW ratio may be a better indicator in fluid imbalance than ECW/TBW ratio. Additionally, alterations in blood anti-inflammatory compounds, lipid metabolites, nucleotide and vitamin metabolites were observed in Se treatment compared with controls. The author raises important issues. The present results are well presented, but the manuscript needs to provide some information. Below are several suggested corrections.
- In the introduction section, authors should explain the reasons for global metabolomic profiles and metabolic pathways measured.
- In the result section, please provide the information for improvement in clinical symptom of BCRL patients
- Line 84, Selenium (Se) instead of Selenium
- Line 86, whole blood Se levels (WBSe) instead of whole blood selenium levels (WBSe).
- Line 99/101, Se instead of selenium
- Line 370, Se stead of SE; Se-rich instead of selenium-rich
- Line 372/374/375/378/392/398…., Se stead of SE
- Expand abbreviations in the result or discussion section. For example, what is LTB4-DMA and PGE3 ? These are not clear?
Round 2
Reviewer 1 Report
Agree to the revised version however two issues were not adressed by the authors but these were acknoledgeded as limitations in the Discussion section.
Author Response
Your valid comments made our results clearer. We really appreciate your efforts and time spent for our paper.